# Cost-effectiveness of human T-cell leukemia virus type 1 (HTLV-1) antenatal screening for prevention of mother-to-child transmission

**Akiko Kowada** *

Department of Occupational Health, Kitasato University Graduate School of Medical Sciences, Sagamihara, Kanagawa, Japan

* kowadaa@gmail.com, kowada.akiko@kitasato-u.ac.jp

## Abstract

### Background

Human T-cell leukemia virus type 1 (HTLV-1) causes adult T-cell leukemia-lymphoma (ATL) and HTLV-1-associated myelopathy-tropical spastic paraparesis (HAM/TSP) with a poor prognosis. This study aimed to evaluate the cost-effectiveness and health impact of HTLV-1 antenatal screening.

### Methodology/Principal findings

A state-transition model was developed for HTLV-1 antenatal screening and no screening over a lifetime horizon from a healthcare payer perspective. A hypothetical cohort of 30-year-old individuals was targeted. The main outcomes were costs, quality-adjusted life-years (QALYs), life expectancy life-years (LYs), incremental cost-effectiveness ratios (ICERs), HTLV-1 carriers, ATL cases, HAM/TSP cases, ATL-associated deaths, and HAM/TSP-associated deaths. The willingness-to-pay (WTP) threshold was set at US$50,000 per QALY gained. In the base-case analysis, HTLV-1 antenatal screening (US$76.85, 24.94766 QALYs, 24.94813 LYs, ICER; US$40,100 per QALY gained) was cost-effective compared with no screening (US$2.18, 24.94580 QALYs, 24.94807 LYs). Cost-effectiveness was sensitive to the maternal HTLV-1 seropositivity rate, HTLV-1 transmission rate with long-term breastfeeding from HTLV-1 seropositive mothers to children, and the cost of the HTLV-1 antibody test. HTLV-1 antenatal screening was cost-effective when the maternal HTLV-1 seropositivity rate was greater than 0.0022 and the cost of the HTLV-1 antibody test was lower than US$94.8. Probabilistic sensitivity analysis using a second-order Monte-Carlo simulation showed that HTLV-1 antenatal screening was 81.1% cost-effective at a WTP threshold of US$50,000 per QALY gained. For 10,517,942 individuals born between 2011 and 2021, HTLV-1 antenatal screening costs US$785 million, increases 19,586 QALYs and 631 LYs, and prevents 125,421 HTLV-1 carriers, 4,405 ATL cases, 3,035 ATL-associated deaths, 67 HAM/TSP cases, and 60 HAM/TSP-associated deaths, compared with no screening over a lifetime.

**Data Availability Statement:** All relevant data are within the manuscript.

**Funding:** The author received no specific funding for this work.

**Competing interests:** The author has declared that no competing interests exist.

## Conclusion/Significance

HTLV-1 antenatal screening is cost-effective and has the potential to reduce ATL and HAM/TSP morbidity and mortality in Japan. The findings strongly support the recommendation for HTLV-1 antenatal screening as a national infection control policy in HTLV-1 high-prevalence countries.

## Author summary

Human T-cell leukemia virus type 1 (HTLV-1) is a carcinogenic retrovirus that causes adult T-cell leukemia-lymphoma (ATL) and HTLV-1-associated myelopathy-tropical spastic paraparesis (HAM/TSP), an unremitting and progressive neurological disorder that presents with spastic paraparesis, neurogenic bladder, sphincter dysfunction, and mild sensory disturbance in the lower extremities. HTLV-1 is endemic in the Southwestern part of Japan, sub-Saharan Africa, South America, the Caribbean area, foci in the Middle East, and Australia-Melanesia. Based on published seroprevalence rates, which are missing or sparse for up to 6/7th of the global population, an estimated 5 to 10 million people worldwide are infected with HTLV-1. Treatment of ATL and HAM/TSP is very difficult and no vaccine is available. HTLV-1 transmission patterns include mother-to-child transmission through breastfeeding, horizontal transmission through sexual intercourse, and direct contact transmission through blood. HTLV-1 antenatal screening is effective to prevent mother-to-child transmission of HTLV-1. The probability of mother-to-child transmission of HTLV-1 through long-term and short-term breastfeeding can be reduced from 20.3% and 7.4% to 2.5% by withholding breastfeeding. However, Japan is the only country in the world that has implemented HTLV-1 antenatal screening. This study demonstrates that HTLV-1 antenatal screening is cost-effective and has the potential to reduce the number of cases and deaths from ATL and HAM/TSP in HTLV-1 high prevalence countries.

## Introduction

Human T-cell leukemia virus type 1 (HTLV-1) is a carcinogenic retrovirus that causes adult T-cell leukemia-lymphoma (ATL) and HTLV-1-associated myelopathy-tropical spastic paraparesis (HAM/TSP), an unremitting and progressive neurological disorder that presents with spastic paraparesis, neurogenic bladder, sphincter dysfunction, and mild sensory disturbance in the lower extremities. These diseases develop after a long incubation period and have a poor prognosis [1–4]. An estimated 5 to 10 million people worldwide are infected with HTLV-1 although data are missing or sparse for up to 6/7th of the global population, based on published seroprevalence rates [5,6]. The regions of HTLV-1 high prevalence are limited to the Southwestern part of Japan, sub-Saharan Africa and South America, the Caribbean area, and foci in the Middle East and Australia-Melanesia [5]. HTLV-1 is transmitted vertically from mother to child through breastfeeding, horizontally through sexual intercourse, and through direct contact via blood [7–9]. HTLV-1 antibody screening for blood donors has been implemented in Australia, Canada, Brazil, Japan, the United Kingdom, Latin America, the Caribbean, and Europe to reduce the risk of HTLV-1 infection through blood product transfusions [10–14].

Japan is the highest-endemic country in the world, where at least 1 million people are estimated to be HTLV-1 carriers [5,8,9]. The HTLV-1 epidemic area used to be limited to the

southern islands of Kyushu and Okinawa, but has recently spread to non-endemic metropolitan areas such as Tokyo and Osaka due to domestic migration [15]. More than 2,800 new cases of HTLV-1 infection are estimated annually in Japan [16]. The number of deaths due to ATL remains close to 1,000 per year [9,15]. Laboratory screening for HTLV-1 infection has been routine practice for blood donors since 1986. The ATL Prevention Program in Nagasaki from 1987 to 2004 revealed a marked reduction of HTLV-1 mother-to-child transmission by withholding breastfeeding for carrier mothers and has recommended avoiding breastfeeding as the most reliable method for mother-to-child transmission prevention [17]. Japan is the first and only country in the world that has implemented nationwide HTLV-1 antenatal screening since 2011 to prevent mother-to-child transmission of HTLV-1 infection [18]. Withholding breastfeeding for mothers with positive HTLV-1 antibody tests at HTLV-1 antenatal screening can reduce the rate of mother-to-child transmission of HTLV-1 from 20.3% for long-term breastfeeding (≧6 months) and 7.4% for short-term breastfeeding (<6 months) down to 2.5% [7]. The Japanese guidelines for obstetrical practice by the Japan Society of Obstetrics and Gynecology and Japan Association of Obstetricians and Gynecologists recommend that all pregnant women from early to mid-term pregnancy, up to about 30 weeks gestation, receive a screening test for HTLV-1 antibodies by particle agglutination or chemiluminescence immunoassay (CLIA) and a confirmatory test by line-blot assay (LIA) or PCR qualitative test [19]. Recently the maternal HTLV-1 seropositivity rate has been gradually decreasing [8,20].

Cost-effectiveness regarding HTLV-1 antenatal screening warrants evaluation as a national infection control policy for HTLV-1.

This study aimed to assess the cost-effectiveness and health impact of HTLV-1 antenatal screening compared with no screening.

## Materials and methods

### Study design

A state-transition model was developed for two intervention strategies: HTLV-1 antenatal screening and no screening. Decision branches were directly connected to one Markov node per intervention strategy and the first event was modeled in a Markov cycle tree. The cycle length was set to one year. A half-cycle correction was applied. Incremental cost-effectiveness ratios (ICERs) were calculated and compared to a willingness-to-pay (WTP) threshold of US $50,000 per quality-adjusted life-year (QALY) gained [21,22].

A hypothetical cohort of 30-year-old individuals born to mothers who had or had not received HTLV-1 antenatal screening was targeted from a healthcare payer perspective with a lifetime horizon.

The main outcome measures were costs, QALYs, life expectancy life-years (LYs), ICERs, HTLV-1 carriers, ATL cases, HAM/TSP cases, ATL-associated deaths, and HAM/TSP-associated deaths.

The model was constructed using TreeAge Pro 2022 (TreeAge Software Inc., Williamstown, Massachusetts). As this was a modeling study with all inputs and parameters derived from the published literature and Japanese statistics, ethics approval was not required.

### Model structure

**HTLV-1 antenatal screening.** For individuals born to mothers with positive HTLV-1 antibody tests at HTLV-1 antenatal screening, withholding breastfeeding reduces the probability of mother-to-child transmission of HTLV-1 from 20.3% for long-term breastfeeding and 7.4% for short-term breastfeeding to 2.5% [7]. Individuals born to mothers with negative HTLV-1 antibody tests have no restrictions on breastfeeding. Children infected with HTLV-1

from their mothers become HTLV-1 carriers. HTLV-1 carriers develop ATL and HAM/TSP depending on the incidence of each disease [23,24]. There are five clinical subtypes of ATL: acute, lymphoma, unfavorable chronic, favorable chronic, and smoldering [1]. Treatment of ATL patients depends on the subtype of ATL. Patients with the aggressive ATL subtypes, acute, lymphoma, and unfavorable chronic types of ATL, receive standard treatment followed by the Japanese practical guidelines for hematological malignancies and the literature [2,25]. Treatment of aggressive ATL subtypes includes allogeneic stem cell transplantation, the only curative therapy available [25]. 44.6% of patients with favorable chronic-type ATL progress to acute-type ATL, and 60% of patients with smoldering-type ATL progress to acute-type ATL [26,27]. Patients with HAM/TSP receive standard treatment followed by the Japanese practical guidelines for HAM [24].

**No screening.** In individuals born to mothers with positive HTLV-1 antibody tests, the probability of being infected with HTLV-1 through breastfeeding is 20.3% for long-term breastfeeding, 7.4% for short-term breastfeeding, and 2.5% for bottle feeding, making them HTLV-1 carriers. Patients with ATL and HAM/TSP receive treatment according to the Japanese practical guidelines for each disease.

## Model inputs

**Clinical probability.** Clinical probabilities were collected using MEDLINE from 2000 to December 2022 (Table 1). To obtain maternal HTLV-1 seropositivity, I assumed the maternal age to be 30 years, which is the average age of first-time pregnant women in Japan [9]. I obtained the maternal HTLV-1 seropositivity rate, HTLV-1 mother-to-child transmission rates with long-term breastfeeding, short-term breastfeeding, and bottle feeding, the incidence of ATL and HAM/TSP in HTLV-1 carriers, the proportion of ATL subtypes in ATL patients, transformation rate from HAM/TSP to ATL, transformation rate from favorable chronic-type and smoldering-type ATL to acute-type ATL, the 4-year survival rates for acute-type and unfavorable chronic-type ATL patients, and the mortality of favorable chronic-type and smoldering-type ATL and HAM/TSP from the literature and Japanese cancer statistics [1,3,4,7,8,23,24,26,27,28,29]. The mortality from other causes was calculated by the adjusted risk of death due to any cause in people with HTLV-1 infection when compared with HTLV-1-negative counterparts [30] and the values obtained from vital statistics [31].

**Cost.** Costs were calculated based on the Japanese medical fee schedule [32] and adjusted to 2021 Japanese yen, using the medical care component of the Japanese consumer price index and converted to 2021 US dollars, using the Organisation for Economic Co-operation and Development (OECD) purchasing power parity rate (US$1 = ¥96.76) (Table 1) [33]. All direct costs were based on a healthcare payer perspective. The cost of the HTLV-1 antibody test was calculated based on the Japanese national fee schedule and medical insurance reimbursement table. The cost consisted of the antibody test by chemiluminescence immunoassay (CLIA) method, the confirmation test by line-blot assay (LIA) method [34], and the immunological test decision fee. Treatment costs for ATL, HAM/TSP, and allogeneic stem cell transplantation were calculated according to the Japanese practical guidelines for hematological malignancies and HAM [2,24,25,32]. All costs were discounted by 3% [35,36].

**Health state utility.** Health status was included to represent nine possible clinical states: (i) HTLV-1 uninfected state, (ii) HTLV-1 carrier, (iii) acute-type ATL, (iv) lymphoma-type ATL, (v) unfavorable chronic-type ATL, (vi) favorable chronic-type ATL, (vii) smoldering-type ATL, (viii) HAM/TSP, and (ix) death (Fig 1). Health state utilities were obtained from the literature [37,38,39] and were calculated using utility weights with values ranging from 1

**Table 1. Input parameters of selected variables.**

| Variable | Baseline value | Sensitivity analysis range | Reference |
|---|---|---|---|
| **Probability** | | | |
| Maternal HTLV-1 seropositivity rate | 0.0027 | 0.001–0.004 | 8 |
| HTLV-1 transmission rate with long-term breastfeeding (≧6 months) from HTLV-1 seropositive mothers to children | 0.203 | 0.1–0.3 | 7 |
| HTLV-1 transmission rate with short-term breastfeeding (<6 months) from HTLV-1 seropositive mothers to children | 0.074 | 0.02–0.1 | 7 |
| HTLV-1 transmission rate with bottle feeding from HTLV-1 seropositive mothers to children | 0.025 | 0.001–0.06 | 7 |
| Proportion of long-term breastfeeding* (≧6 months) | 0.374 | 0.3–0.5 | 28,29 |
| Proportion of short-term breastfeeding* (<6 months) | 0.591 | 0.5–0.7 | 28,29 |
| Proportion of bottle feeding | 0.035 | 0–0.1 | 28,29 |
| Annual ATL rate in HTLV-1 carriers | 0.000976 | 0.0005–0.0015 | 23 |
| Proportion of acute-type ATL in ATL patients | 0.508 | 0.477–0.539 | 1 |
| Proportion of lymphoma-type ATL in ATL patients | 0.249 | 0.222–0.276 | 1 |
| Proportion of unfavorable chronic-type ATL in ATL patients | 0.094 | 0.076–0.112 | 1 |
| Proportion of favorable chronic-type ATL in ATL patients | 0.044 | 0.031–0.057 | 1 |
| Proportion of smoldering-type ATL in ATL patients | 0.105 | 0.086–0.124 | 1 |
| Transformation rate from favorable chronic-type ATL to acute-type ATL | 0.446 | 0.4–0.8 | 26,27 |
| Transformation rate from smoldering-type ATL to acute-type ATL | 0.6 | 0.4–0.8 | 26,27 |
| HAM/TSP rate in HTLV-1 carriers | 0.000033 | 0.000022–0.000044 | 24 |
| Transformation rate from HAM/TSP to ATL | 0.00381 | 0.00343–0.00419 | 3 |
| 4-year acute-type ATL survival rate (%) | 16.8 | 6.7–26.9 | 1 |
| 4-year lymphoma-type ATL survival rate (%) | 19.6 | 9.7–29.5 | 1 |
| 4-year unfavorable chronic-type ATL survival rate (%) | 26.6 | 16.8–36.0 | 1 |
| Mortality of favorable chronic-type and smoldering-type ATL | 0.079 | 0.06–0.12 | 26,27 |
| Mortality of HAM/TSP | 0.024 | 0.015–0.033 | 4 |
| Adjusted risk of death due to any cause in people with HTLV-1 when compared with HTLV-1-negative counterparts | 1.57 | 1.37–1.80 | 30 |
| **Cost, US$ (US$1 = ¥ 96.76)** | | | |
| HTLV-1 antibody initial and confirmation test | 76.4 | 57.3–95.5 | 32,34 |
| Treatment for ATL | 108,581 | 81,436–135,725 | 2,25,32 |
| Treatment for HAM/TSP | 6,700 | 5,025–8,375 | 24,32 |
| **Utility** | | | |
| HTLV-1 uninfected state | 1 | 0.7–1 | 37,38,39 |
| HTLV-1 carrier | 0.712 | 0.684–1 | |
| Acute-type ATL | 0.67 | 0.58–0.76 | |
| Lymphoma-type ATL | 0.67 | 0.58–0.76 | |
| Unfavorable chronic-type ATL | 0.67 | 0.58–0.76 | |
| Favorable chronic-type ATL | 0.69 | 0.60–0.78 | |
| Smoldering-type ATL | 0.7 | 0.61–0.79 | |
| HAM/TSP | 0.299 | 0.271–0.327 | |
| Death | 0 | N/A | |

HTLV-1, human T cell leukemia virus 1; ATL, Adult T-cell leukemia-lymphoma; HAM/TSP, HTLV-1-associated myelopathy-tropical spastic paraparesis; N/A, not applicable

* Breastfeeding includes a mixture of breastfeeding and bottle feeding.

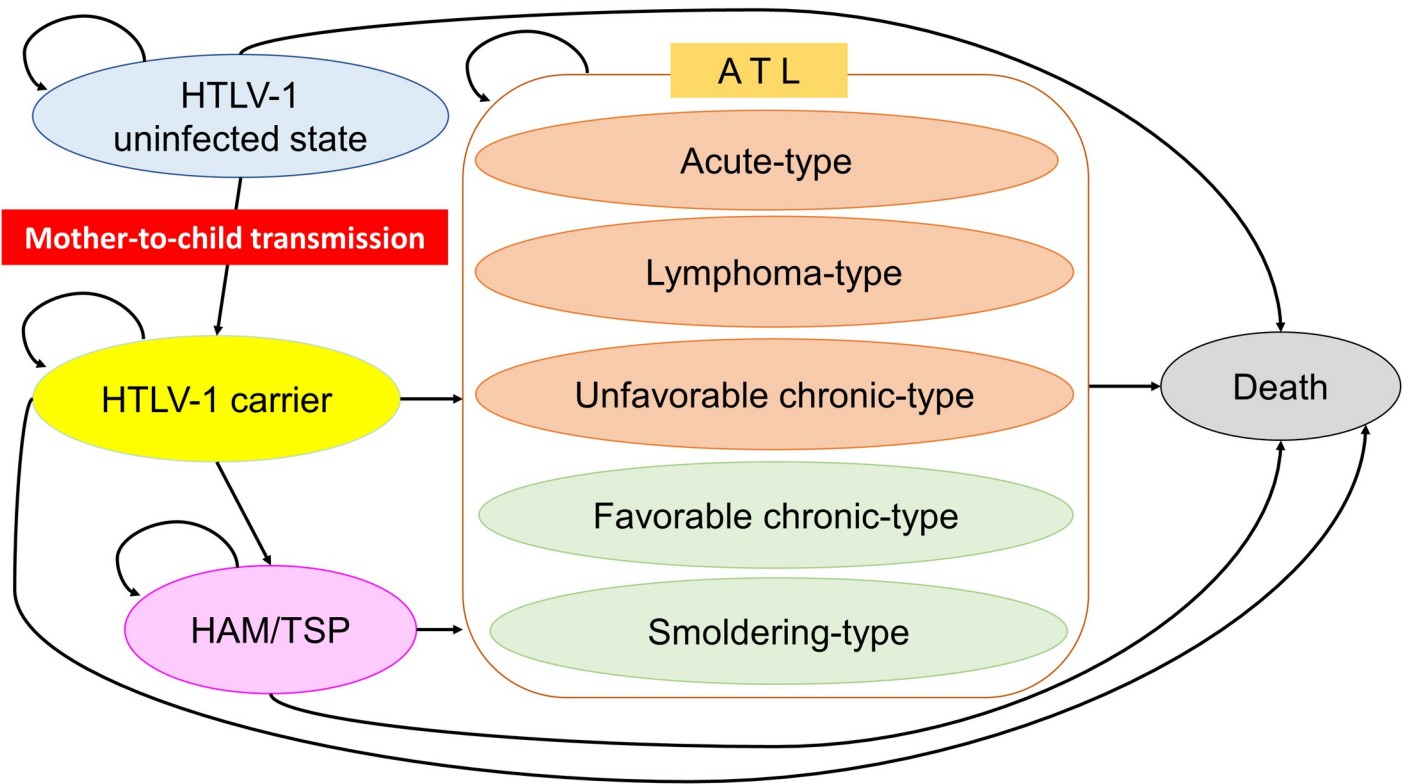

**Fig 1. Simplified schematic depiction of a state transition diagram.** The health states in the model are shown to be oval. In a yearly model cycle, transition paths occur between the health states and other health states, as represented by the arrows. HTLV-1, human T-cell leukemia virus type 1; ATL, adult T-cell leukemia-lymphoma; HAM/TSP, HTLV-1-associated myelopathy-tropical spastic paraparesis.

(healthy) to 0 (death) (Table 1). The annual discounting of health state utilities was set at a rate of 3% [35,36].

## Sensitivity analysis

To determine which strategy would be more cost-effective if one variable was tested over the widest possible range, holding all other variables constant, one-way sensitivity analyses were conducted on variables such as the maternal HTLV-1 seropositivity rate, the incidence of ATL and HAM/TSP among HTLV-1 carriers, the mortality of ATL and HAM/TSP, the cost of HTLV-1 antibody test, treatment cost of ATL and HAM/TSP, and health utilities (Table 1). A two-way sensitivity analysis was conducted on the maternal HTLV-1 seropositivity rate and the proportion of long-term breastfeeding. To assess the impact of model uncertainty on the base case estimates, the probabilistic sensitivity analysis using a second-order Monte-Carlo simulation over 1000 trials was also performed. The uncertainty had a beta distribution for probability and accuracy, and a gamma distribution for cost.

## Markov cohort analysis

The Markov cohort analyses determined the cumulative lifetime probability of ATL cases and deaths, HAM/TSP cases and deaths, and HTLV-1 carriers in HTLV-1 antenatal screening and no screening. The cumulative lifetime number of ATL cases and deaths, HAM/TSP cases and

deaths, and HTLV-1 carriers were obtained by multiplying the cumulative lifetime probabilities of the two strategies by the total number of births from 2011 to 2021 (10,517,942) [31].

## Results

### Base-case analysis

HTLV-1 antenatal screening (US$76.85, 24.94766 QALYs, ICER; US$40,100 per QALY gained, 24.94813 LYs, ICER; US$1,245,303 per LY gained) was cost-effective compared to no screening (US$2.18, 24.94580 QALYs, 24.94807 LYs) (Table 2).

### Sensitivity analyses

ICER tornado diagram showed that HTLV-1 antenatal screening is cost-effective compared with no screening at a WTP threshold of US$50,000 per QALY gained when the maternal HTLV-1 seropositivity rate is greater than 0.0022, HTLV-1 transmission rate with long-term breastfeeding from HTLV-1 seropositive mothers to children is greater than 0.154, HTLV-1 transmission rate with short-term breastfeeding from HTLV-1 seropositive mothers to children is greater than 0.043, HTLV-1 transmission rate with bottle feeding from HTLV-1 seropositive mothers to children is lower than 0.044, the health utility value of HTLV-1 carriers is lower than 0.77, and the cost of HTLV-1 antibody test is lower than US$94.8 (Fig 2). The two-way sensitivity analysis revealed that HTLV-1 antenatal screening is more cost-effective the higher the maternal HTLV-1 seropositivity rate and the higher the proportion of long-term breastfeeding (Fig 3). In the probabilistic sensitivity analysis using a second-order Monte-Carlo simulation for 1000 trials, the acceptability curve showed that HTLV-1 antenatal screening is 81.1% cost-effective at a WTP threshold of US$50,000 per QALY gained (Fig 4). The incremental cost-effectiveness scatterplot showed that HTLV-1 antenatal screening is dominant for 811 trials to no screening in 1000 trials at a WTP threshold of US$50,000 per QALY gained (Fig 5).

### Cumulative lifetime economic and health outcomes

For 10,517,942 individuals born between 2011 and 2021, HTLV-1 antenatal screening costs US$785 million, increases 19,586 QALYs and 631 LYs, and prevents 125,421 HTLV-1 carriers, 4,405 ATL cases, 67 HAM/TSP cases, 3,035 ATL-associated deaths, and 60 HAM/TSP-associated deaths, compared with no screening over a lifetime (Table 3).

## Discussion

This study demonstrated that HTLV-1 antenatal screening is cost-effective compared with no screening in Japan. The main reason for the superior cost-effectiveness of HTLV-1 antenatal screening is that withholding breastfeeding for HTLV-1 seropositive mothers significantly

**Table 2. Base-case analysis.**

| Strategy | Cost, US$ | Incremental cost, US$ | Effectiveness, QALYs | Incremental QALYs | ICER, US$/QALY gained | Effectiveness, LYs | Incremental LYs | ICER, US$/LY gained |
|---|---|---|---|---|---|---|---|---|
| No screening | 2.18 | - | 24.94580 | - | - | 24.94807 | - | - |
| HTLV-1 antenatal screening | 76.85 | 74.67 | 24.94766 | 0.00186 | 40,100 | 24.94813 | 0.00006 | 1,245,303 |

HTLV-1, human T cell leukemia virus 1; QALY, quality-adjusted life-year; LY, life expectancy life-year; ICER, incremental cost-effectiveness ratio

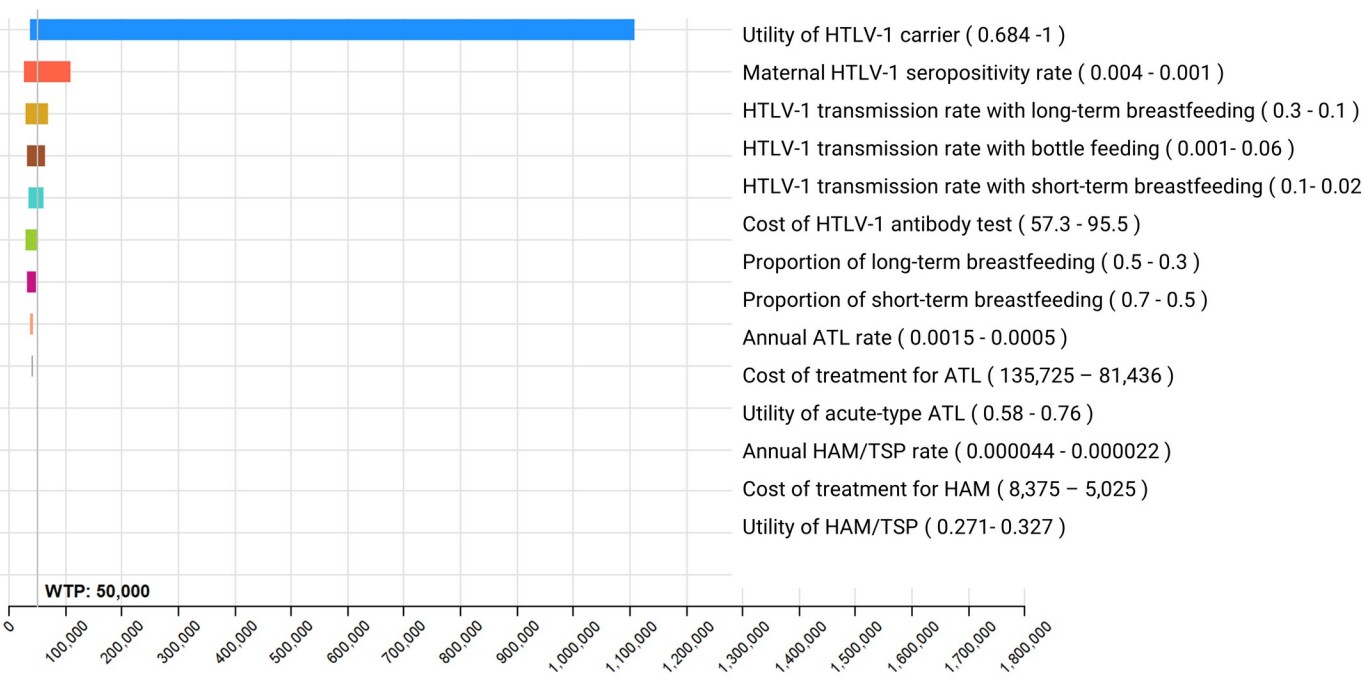

**Fig 2. ICER tornado diagram for HTLV-1 antenatal screening versus no screening.** HTLV-1 antenatal screening is cost-effective compared with no screening at a WTP threshold of US$50,000 per QALY gained when the maternal HTLV-1 seropositivity rate is greater than 0.0022, HTLV-1 transmission rate with long-term breastfeeding from HTLV-1 seropositive mothers to children is greater than 0.154, HTLV-1 transmission rate with short-term breastfeeding from HTLV-1 seropositive mothers to children is greater than 0.043, HTLV-1 transmission rate with bottle feeding from HTLV-1 seropositive mothers to children is lower than 0.044, the health utility value of HTLV-1 carriers is lower than 0.77, and the cost of HTLV-1 antibody test is lower than US$94.8. ICER, incremental cost-effectiveness ratio; WTP, willingness-to-pay; QALY, quality-adjusted life-year; HTLV-1, human T cell leukemia virus 1; ATL, adult T-cell leukemia-lymphoma; HAM/TSP, HTLV-1-associated myelopathy-tropical spastic paraparesis.

reduces the mother-to-child transmission rate of HTLV-1 infection, resulting in fewer HTLV-1 carriers and subsequently fewer ATL and HAM/TSP cases and deaths.

The study showed that the lifetime per person cost of HTLV-1 antenatal screening was US $74.67 higher than that of no screening. However, further cost savings may be expected in the future by promoting low-cost HTLV-1 antenatal screening through high-volume screening worldwide.

To the best of my knowledge, this is the first modelling study in HTLV-1 high-prevalence countries to evaluate the cost-effectiveness and health impact of HTLV-1 antenatal screening. Malik et al suggested, based on a modelling study, that HTLV-1 antenatal screening meets the cost-efficacy standards in the UK, one of the HTLV-1 low-prevalence countries [40]. This study demonstrated that HTLV-1 antenatal screening is cost-effective in Japan, one of the HTLV-1 high-prevalence countries.

The main HTLV-1 highly endemic regions are localized in the world: the Southwestern part of Japan, sub-Saharan Africa and South America, the Caribbean area, and foci in the Middle East and Australia-Melanesia [5]. Djuicy et al demonstrated that rural adult populations in Gabon are highly endemic to HTLV-1, with an overall prevalence of 8.7% [41]. Paiva et al reported that high maternal HTLV-1 proviral load and breastfeeding beyond 12 months were

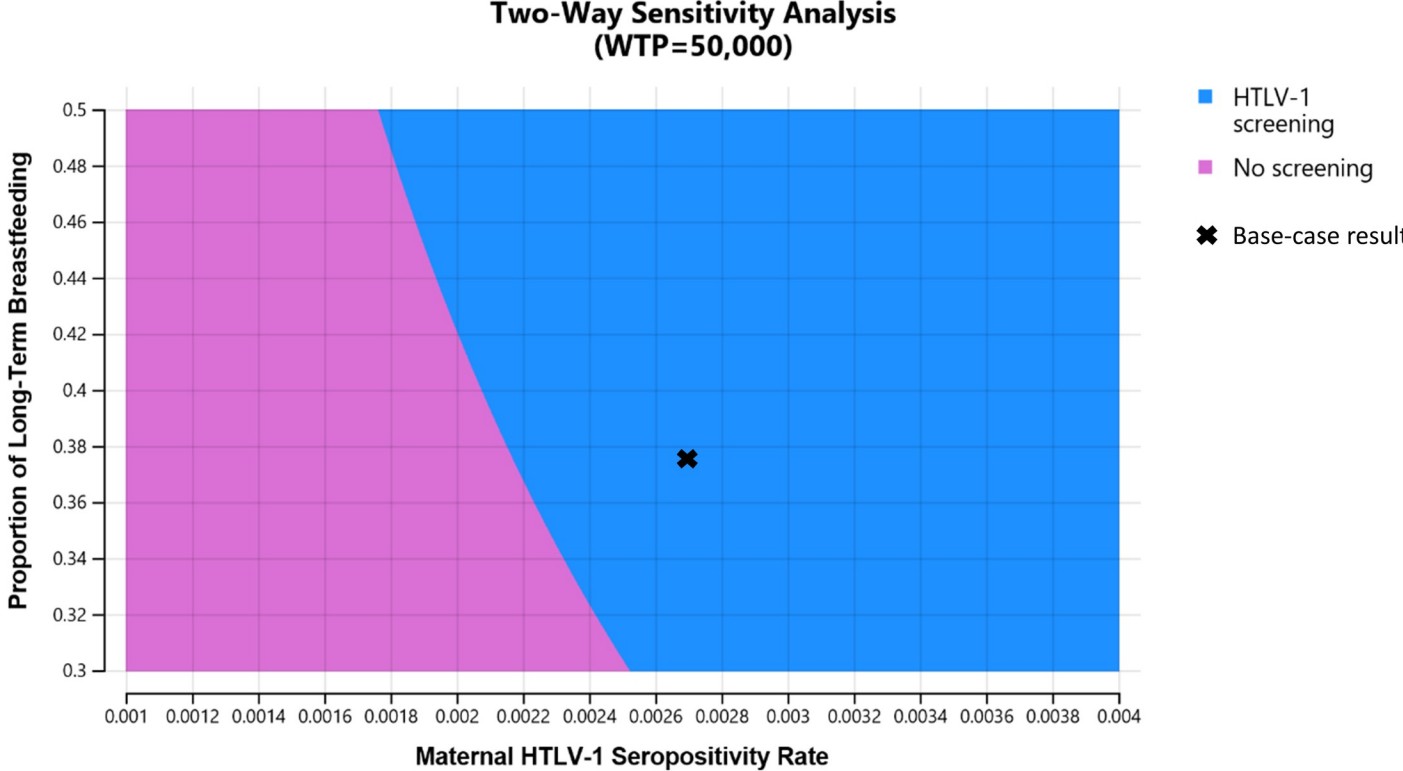

**Fig 3. Two-way sensitivity analysis for the maternal HTLV-1 seropositivity rate and the proportion of long-term breastfeeding.** HTLV-1 antenatal screening is optimal in the blue region which includes the base-case result (x). No screening is optimal in the orchid region. This figure shows that HTLV-1 antenatal screening is more cost-effective the higher the maternal HTLV-1 seropositivity rate and the higher the proportion of long-term breastfeeding. WTP, willingness-to-pay; HTLV-1, human T cell leukemia virus 1.

independently associated with mother-to-child transmission of HTLV-1 infection, highlighting the need for both antenatal HTLV-1 screening and advising mothers on breastfeeding in Brazil [42]. Ngoma et al found that the pooled HTLV-1 seroprevalence was 3.19% (95% CI 2.36–4.12%) in sub-Saharan Africa, indicating the need to implement effective prevention programs and interventions in sub-Saharan Africa [43]. In these HTLV-1 high-prevalence countries, HTLV-1 antenatal screening may be more cost-effective the higher the maternal HTLV-1 seropositivity rate and the higher the proportion of long-term breastfeeding, based on the findings. Policy makers in high-endemic countries could consider implementing HTLV-1 antenatal screening to prevent mother-to-child transmission as a cost-effective strategy for national HTLV-1 infection control.

Recently, horizontal transmission of HTLV-1 through sexual intercourse has also become an important route of HTLV-1 infection [44]. A large-scale educational campaign that includes condom use and avoidance of high-risk sexual behaviors would be effective in preventing sexual transmission of HTLV-1 just as an HIV campaign has been effective [45,46]. It is important to raise awareness about HTLV-1 infection and prevention not only in endemic areas but also in all regions of the country.

Unfortunately, no vaccine is currently available, likely due to the low interest of pharmaceutical companies associated with the restricted markets in industrialized countries [47]. However, HTLV-1 causes serious diseases, ATL and HAM/TSP, and thus places in the same category of viruses for which efficient vaccines are made and used. Furthermore, there are

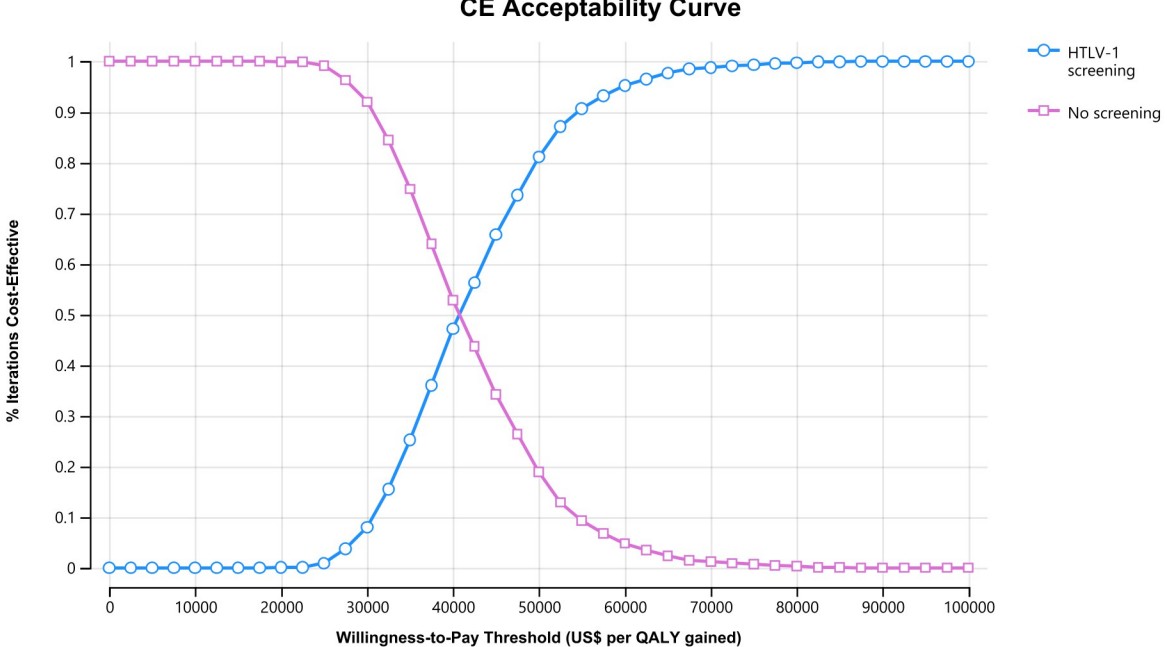

**Fig 4. Cost-effectiveness acceptability curve.** The probabilistic sensitivity analysis analyzes 1000 simulations of the model in which input parameters are randomly varied across pre-specified statistical distributions. The x-axis represents the WTP threshold. The acceptability curve showed that HTLV-1 antenatal screening is 81.1% cost-effective at a WTP threshold of US$50,000 per QALY gained. CE, cost-effectiveness; QALY, quality-adjusted life-year; WTP, willingness-to-pay; HTLV-1, human T cell leukemia virus 1.

factors favoring the feasibility of the vaccine against HTLV-1: HTLV-1 has a relatively low antigenic variability, natural immunity occurs in humans, and experimental vaccination with the envelope antigen is successful in animal models. Vaccines against HTLV-1 would have important public health value in oncology, neurology, and AIDS, and are expected to play a pioneering role in the field of oncology, neurology, and AIDS [48]. In the future, vaccines against HTLV-1 should be developed and widely served to prevent transmission of HTLV-1 from mother to child and between sexual partners.

There are negative impacts of not breastfeeding. For mothers, it increases the long-term risk of cancer (breast, ovarian, endometrium), endometriosis, diabetes, osteoporosis, hypertension, cardiovascular diseases, metabolic syndrome, rheumatoid arthritis, Alzheimer disease, and multiple sclerosis. For infants, it increases the risk of otitis media, upper and lower respiratory tract infection, asthma, RSV bronchiolitis, atopic dermatitis, gastroenteritis, inflammatory bowel disease, diabetes, leukemia, and sudden infant death syndrome [49,50].

This study has several limitations. First, the cost of health supervision and counseling for pregnant women with confirmed HTLV-1 infection at HTLV-1 antenatal screening was not taken into account. Second, the costs of hospitalization, rehabilitation, and complications from treatment of ATL and HAM/TSP were not included. Third, HTLV-1 horizontal transmission rate was not taken into account in the model. The HTLV-1 horizontal transmission rate from males to females is greater than the HTLV-1 horizontal transmission rate from females to males [16]. Fourth, nonmedical indirect costs, such as lost productivity, work absenteeism, and income loss, were not calculated in this study. Fifth, the model did not consider the disadvantages such as depression and anxiety felt by HTLV-1-positive mothers and their families, and the influence on their children of withholding breastfeeding. Sixth, the adverse

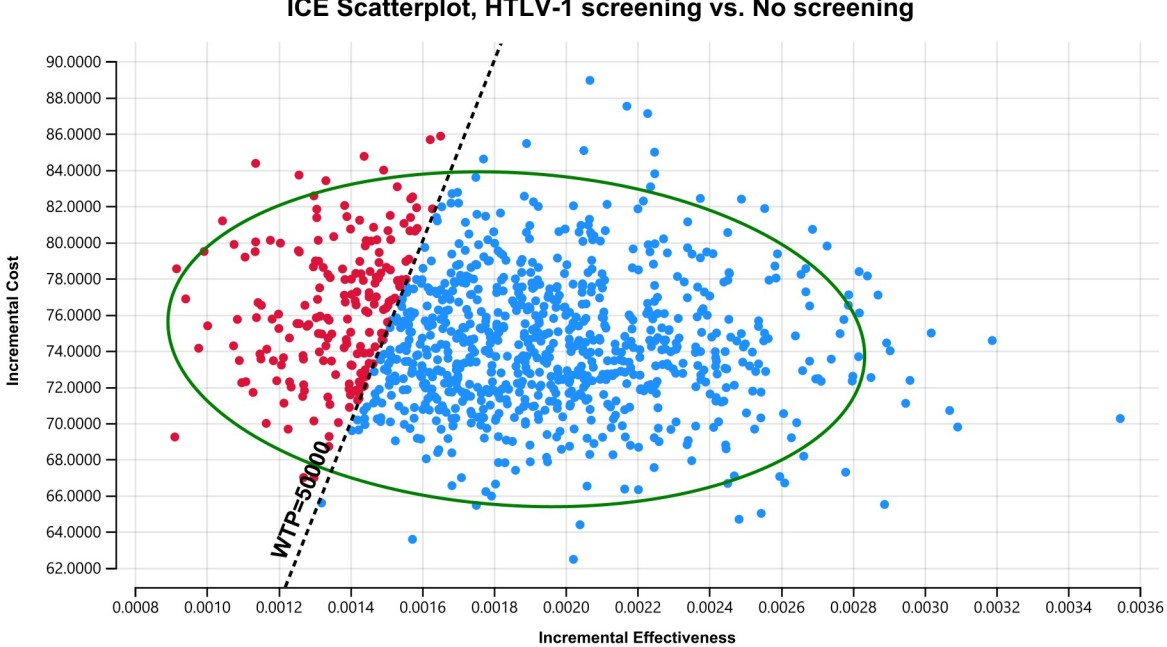

**Fig 5. ICE scatterplot with a 95% confidence ellipse at a WTP threshold of US$50,000 per QALY gained.** Each dot represents a single simulation for a total of 1000 simulations. The ICE scatterplot showed that HTLV-1 antenatal screening is dominant in 811 trials to no screening in 1000 trials. ICE, incremental cost-effectiveness; QALY, quality-adjusted life-year; WTP, willingness-to-pay; HTLV-1, human T cell leukemia virus 1.

health effects on mothers and infants of withholding breastfeeding were not considered in the model. Seventh, twins were not taken into account in the analysis. Eighth, the model didn't account for other HTLV-1-associated diseases such as HTLV-1-associated uveitis, infective dermatitis, bronchiectasis, bronchitis, bronchiolitis, seborrheic dermatitis, Sjögren's syndrome, rheumatoid arthritis, fibromyalgia, and ulcerative colitis. Finally, there are different

**Table 3. Economic and health outcomes of HTLV-1 antenatal screening vs no screening.**

| Outcome | HTLV-1 antenatal screening | | No screening | | Difference, HTLV-1 antenatal screening vs no screening | |
|---|---|---|---|---|---|---|
| | Per person | Per 10,517,942 persons* | Per person | Per 10,517,942 persons* | Per person | Per 10,517,942 persons* |
| Cumulative lifetime cost, US$ | 76.85 | 808,322,795 | 2.18 | 22,910,660 | 74.67 | 785,412,136 |
| Cumulative lifetime QALYs | 24.94766 | 262,398,045 | 24.94580 | 262,378,459 | 0.00186 | 19,586 |
| Cumulative lifetime LYs | 24.94813 | 262,403,006 | 24.94807 | 262,402,375 | 0.00006 | 631 |
| Cumulative lifetime HTLV-1 carriers | 0.0031206 | 32,822 | 0.0150451 | 158,244 | -0.0119245 | -125,421 |
| Cumulative lifetime ATL cases | 0.0001096 | 1,153 | 0.0005284 | 5,558 | -0.0004188 | -4,405 |
| Cumulative lifetime ATL-associated deaths | 0.0000755 | 794 | 0.0003641 | 3,830 | -0.0002886 | -3,035 |
| Cumulative lifetime HAM/TSP cases | 0.0000017 | 18 | 0.0000081 | 86 | -0.0000064 | -67 |
| Cumulative lifetime HAM/TSP-associated deaths | 0.0000015 | 16 | 0.0000072 | 75 | -0.0000057 | -60 |

HTLV-1, human T cell leukemia virus 1; QALY, quality-adjusted life-year; LY, life expectancy life-year; ATL, Adult T-cell leukemia-lymphoma; HAM/TSP, HTLV-1-associated myelopathy-tropical spastic paraparesis

*Between 2011 and 2021, 10,517,942 babies were born in Japan.

costs and medical systems in each country. Further cost-effectiveness studies by the variance of each country are needed.

## Conclusion

This study demonstrated that HTLV-1 antenatal screening is cost-effective and has the potential to reduce ATL and HAM/TSP morbidity and mortality in Japan. The findings strongly support the recommendation for HTLV-1 antenatal screening as a national infection control policy in HTLV-1 high-prevalence countries.

## Author Contributions

**Conceptualization:** Akiko Kowada.

**Data curation:** Akiko Kowada.

**Formal analysis:** Akiko Kowada.

**Investigation:** Akiko Kowada.

**Methodology:** Akiko Kowada.

**Project administration:** Akiko Kowada.

**Resources:** Akiko Kowada.

**Software:** Akiko Kowada.

**Validation:** Akiko Kowada.

**Visualization:** Akiko Kowada.

**Writing – original draft:** Akiko Kowada.

**Writing – review & editing:** Akiko Kowada.

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
