## [Decision Letter · Decision Letter 0]

10 Oct 2022

Dear Dr. Kowada,

Thank you very much for submitting your manuscript "Cost-effectiveness and health impact of HTLV-1 antenatal screening for prevention of mother-to-child transmission" for consideration at PLOS Neglected Tropical Diseases. As with all papers reviewed by the journal, your manuscript was reviewed by members of the editorial board and by several independent reviewers. In light of the reviews (below this email), we would like to invite the resubmission of a significantly-revised version that takes into account the reviewers' comments. 

We cannot make any decision about publication until we have seen the revised manuscript and your response to the reviewers' comments. Your revised manuscript is also likely to be sent to reviewers for further evaluation.

Sincerely,

Eugenia Corrales-Aguilar

Section Editor

Eugenia Corrales-Aguilar

Section Editor

Reviewer's Responses to Questions

**Key Review Criteria Required for Acceptance?**

**Methods**

-Are the objectives of the study clearly articulated with a clear testable hypothesis stated?

-Is the study design appropriate to address the stated objectives?

-Is the population clearly described and appropriate for the hypothesis being tested?

-Is the sample size sufficient to ensure adequate power to address the hypothesis being tested?

-Were correct statistical analysis used to support conclusions?

-Are there concerns about ethical or regulatory requirements being met?

Reviewer #1: Methods need clarification as per comments on the Summary and General Comments session.

Reviewer #2: (No Response)

**Results**

-Does the analysis presented match the analysis plan?

-Are the results clearly and completely presented?

-Are the figures (Tables, Images) of sufficient quality for clarity?

Reviewer #1: Please see comments on Summary and General Comments session.

Reviewer #2: (No Response)

**Conclusions**

-Are the conclusions supported by the data presented?

-Are the limitations of analysis clearly described?

-Do the authors discuss how these data can be helpful to advance our understanding of the topic under study?

-Is public health relevance addressed?

Reviewer #1: Please see comments on Summary and General Comments session.

Reviewer #2: (No Response)

**Editorial and Data Presentation Modifications?**

Reviewer #1: Minor Comments.

1. The estimate of 20 million people infected with HTLV-1 worldwide is from de The & Bomford(3), not Gessain & Cassar(4)– who estimated 5 – 10 million based on published seroprevalence but acknowledged that data pertinent to 6/7th of the global population were missing or sparse.

2. Regions of high prevalence include Iran and Central Australia – this is mentioned in the discussion but omitted from the introduction.

3. In the author summary an explanation of the type of disease would be helpful for HAM/TSP (as is given for ATL).

4. Line 123 countries which screen – please provide a reference.

5. Line 365 – How do you know that the proposed large scale education campaign will be effective in preventing sexual transmission? Perhaps compare to HIV campaign and reference.

6. The term “we” is used in the manuscript but there is only one author. In which case either use I or write in the third person.

Reviewer #2: (No Response)

**Summary and General Comments**

Reviewer #1: The addition of a cost effectiveness study from Japan on antenatal screening for prevention of mother-to-child HTLV-1 transmission is welcome. Indeed, it is interesting that no such study has previously been published either leading up to the decision to extend HTLV antenatal screening nationally circa 2011 or since. The only previous study, from the UK– a low prevalence country – concluding that even in the UK screening could be cost effective(1). The paper by A. Kowada concludes that antenatal screening in Japan meets the cost criteria for screening based on provider willingness to pay values of US$ 50,000 or 100,000 per QALY. These are higher than in the UK

Major Comments.

1. The life-time risk of HAM/TSP cited in the introduction is 0.18 – 1.8% (although higher rates have been reported. Which rate was used for this study?

2. The life-time risk of ATL is cited in the introduction to be 5%. Was the association of ATL risk with acquisition of HTLV-1 in early life considered in the analysis?

3. Why were the thresholds $50 and $100,000 chosen? What is the accepted cost per QALY in Japan?

4. In the methods it is not clear what “other causes” of mortality were considered in the model. In the results on HAM/TSP and ATL cases are presented. Please clarify. Also, what impact does the adj Mortality Rate of 1.57 as reported by Schierhout(2) have on the cost effectiveness? Noting that this cannot be attributed to deaths from ATL or HAM

5. In the model structure the assumption is that all mothers breast-feed. Is this the case in Japan and if so for how long?

6. What is the negative impact (QALY cost) of not breast-feeding?

7. The effectiveness of the intervention is given as reducing transmission from 20% to 3% but the rate of transmission is determined by the duration of breast feeding. This should be taken into account in the model.

8. The cost of the antibody screening is given as US$ 76.4 per person inclusive of confirmatory test. How was this calculated? It is taken from a table of medical fees but high-volume screening is likely to be cheaper in terms of the reagents and in personnel as it is added to other screening assays. Has this been considered?

9. The impact of sexual transmission was considered. What was estimate of life-time risk that a person infected through in early life would transmit HTLV-1 through sexual intercourse? How was it calculated? Was this different for males and females?

10. In the section on cumulative lifetime health effects the number of cases of ATL and HAM prevented seem small for ATL with an expected ATL rate of 2.8% only and very small for HAM with a lifetime risk of just 0.15%. A detailed explanation of these rates is required for the reader to understand the findings. 

11. In the analysis how are twins accounted for?

12. In Figure 2C the pivot point for effectiveness seems to be at about $25,000 per QALY). Why consider $100,000 per QALY?

13. Table 3 is confusing with cumulative life-time cost of HTLV-1 antenatal screening given as US$ 866 million and various sums presented as reduction regardless of whether these are losses or gains.

14. Furthermore the reduction in cases in Table 3 requires explanation. Why is the reduction in disease not at the same as the reduction in cases of infection?

15. In the discussion reference is made to the implementation of a national antenatal screening programme in 2011 and that recently maternal seroprevalence has been decreasing. Clearly an antenatal screening programme from 2011 cannot have impacted maternal seroprevalence already especially as the average age of mothers in Japan is 30 years. Comment should be made on the extent of screening in Japan in the preceding years noting the introduction of HTLV antenatal screening in Nagasaki was early as the late 1980s?

16. In the comment on depression and anxiety whilst acknowledging that all diagnostic tests and screening programmes are inevitably associated with a degree of anxiety the current statement implies that this would be particularly a feature of HTLV screening and does not take into account the depression and anxiety that occurs when parents discover that they were not offered testing and interventions for a preventable infection causing cancer and neurological disease (as HTLV-1) and see these occurring in the children (even as adults). It is important to discuss both elements since the psychological aspects are not included – not just the potential negative aspects.

1. Malik B, Taylor GP. Can we reduce the incidence of adult T-cell leukaemia/lymphoma? Cost-effectiveness of human T-lymphotropic virus type 1 (HTLV-1) antenatal screening in the United Kingdom. Br J Haematol. 2018;184:1040-3.

2. Schierhout GM, S; Gessain, A; Einsiedel, L; Martinello, M, Kaldor, J. The association between HTLV-1 infection and adverse health outcomes:a systematic review and meta-analysis of epidemiologic studies. Lancet Infect Dis. 2019.

3. de The G, Bomford R. An HTLV-I vaccine: why, how, for whom? AIDS Res Hum Retrovirol. 1993;9(5):381-6.

4. Gessain A, Cassar O. Epidemiological Aspects and World Distribution of HTLV-1 Infection. Front Microbiol. 2012;3:388.

Reviewer #2: 1. As this paper is written, it appears to have an overwhelmingly major flaw: It seems to assume that prevention of an HTLV-1 infection increases the utility of the person in question in by far most cases from 0.712 to 1. The author appears to assume that a person who is not an HTLV-1 carrier, is in perfect health and has a utility of 1. That assumption is incorrect because there are many other factors and conditions than HTLV-1 that can cause less than perfect health. Correction would greatly reduce the number of QALYs gained by screening, possibly by a factor greater than 10, so that cost-effectiveness of screening would change from well within what is generally considered cost-effective, to well outside of that. Correction would therefore reverse the main conclusion of the paper and require a major revision of the text and tables.

Some comments of less importance than the above:

2. The author mentions several times “more cost-effective” while the paper only compares screening with not-screening, therefore there is only one cost-effectiveness ratio at play, so that there is no comparison (“more”) of cost-effectiveness ratios possible.

3. The text seems often unclear about whether a number mentioned concerns incidence or prevalence. E.g. line 78 seems to describe prevalence, while line 111 seems to quote the same number but describes it as incidence.

4. At line 155: are those individuals all pregnant women? If so, please say so.

5. At line 171: as I read the paper, compliance rate does not affect the estimated cost-effectiveness ratio

6. At line 191 under the heading ‘no screening’: is the probability of sexual transmission not considered in the screening arm?

7. It is generally left unclear how sexual transmission affects the cost-effectiveness of screening. Does it have any influence on the presented estimates?

8. Does direct transmission play any role in this model?

9. The one-way sensitivity analysis does not seem to just involve changing one parameter at the time (with results usually presented in a tornado diagram), but to involve only one parameter altogether.

10. At line 207: Reference 20 does not describe test specificity as 100%. Furthermore, the false positivity rates mentioned there, would cause that around half of test positives are false positives.

11. At line 336: This paper does not demonstrate that screening reduces morbidity and mortality but rather, makes literature based quantitative assumptions concerning effects on morbidity and mortality in order to estimate cost-effectiveness.

12. At line 379: The paper leaves unclear how a gender difference in horizontal transmission affects the cost-effectiveness of screening.

13. At lines 386-389: As mentioned under 1., the paper seems to assume that HTLV-1 carriers have a much lower utility than non-carriers therefore it seems incorrect to state that other diseases than ATL and HAM/TSP were not taken into account, on the contrary, it seems that their effects are gravely overestimated.

PLOS authors have the option to publish the peer review history of their article (what does this mean?). If published, this will include your full peer review and any attached files.

Reviewer #1: No

Reviewer #2: No
---

## [Decision Letter · Decision Letter 1]

6 Dec 2022

Dear Dr. Kowada,

Thank you very much for submitting your manuscript "Cost-effectiveness and health impact of HTLV-1 antenatal screening for prevention of mother-to-child transmission" for consideration at PLOS Neglected Tropical Diseases. As with all papers reviewed by the journal, your manuscript was reviewed by members of the editorial board and by several independent reviewers. The reviewers appreciated the attention to an important topic. Based on the reviews, we are likely to accept this manuscript for publication, providing that you modify the manuscript according to the review recommendations. 

Thank you for addressing point by point the reviewers' comments. This has been most helpful. Further clarification of how onward sexual transmission was incorporated into the model is required as well as how the contribution of sexually acquired transmission to the total number of cases of HTLV-1 in Japan has been factored into the incidence data. See the points below and the further comments from Reviewer 1.

Line 131 – are 4000 infections in adults and adolescents diagnosed each year or is this the estimated number of new infections per year?

Line 217 and Table 1– the horizontal infection rate is 0.000046 on the 2016 paper of HTLV-1 incidence in Japanese blood donors. This is the infection rate but not the transmission rate. A carrier infected in early life will have potentially more opportunity to transmit than a carrier who does not become infected until they become sexually active. Please also confirm that the transmission rate when incorporated into the model describes the life-time risk of transmission following early life infection and not the risk per 100,000 person years. This is not clear at the moment. I note too that the work of Satake et al has recently been updated in J Clin Virol 2022.

(The incidence density was significantly higher in women (6·88 per 100 000 person-years; 95% CI 6·17-7·66) than in men (2·29 per 100 000 person-years; 95% CI 1·99-2·62; p<0·0001). 

The number of seroconversions per 100,000 person-years was 1.54 for men and 4.21 for women – satake et al J Clin Virol . 2022 Dec;157:105324. doi: 0.1016/j.jcv.2022.105324.

Although the seroconversions per 100,000 person years is lower than in the 2016 report the sero-conversion rate amongst adolescents and Young adults had increased.

Line 307 – “The sexual transmission rate didn’t affect the cost-effectiveness of screening”. Do you mean that the number of transmissions attributed to early life infection was so low that if did not increase the cost-effectiveness of screening. How many sexual transmissions would have been prevented by the avoidance of mother-to-child transmission. This should be added to the section on cumulative lifetime health effects.

In this section the estimate on ATL-associated deaths is 64% of the total of ATL cases whereas the data input was that the 4 year survival of 16%. Why is the mortality rate in the model so low. On the other hand 71 HAM associated deaths were prevented out of 177 cases – a 40% mortality. The suggestion in the text is that these numbers may be wrong. What is their contribution to the cost-effectiveness?

In the response to Reviewer one you state Yes, the association of ATL risk with acquisition of HTLV-1 in early life was considered in the analysis. In the model, HTLV-1 is transmitted mainly through mother-to-child transmission (MTCT), even if horizontal transmission through sexual intercourse is being considered. However, the data used for ATL risk was the published incidence rate – which would include all persons infected with HTLV-1 regardless of the age of infection. Whilst ATL has been reported following infection in adult life a strong association with infection in early life remains. The relative contribution of infection in early v adult life is therefore likely to impact significantly in the model. What proportion of all HTLV-1 infection in adults in Japan did you consider to be related to mother-to-child transmission? The increasing rates of infection with increasing age, especially among females, indicate a considerable infection after early life. 

In your response to point 16 Reviewer 1 you have stated that anxiety related to screening would be a particular feature of HTLV-1 screening. It is not clear that this was your intention. If not, please amend. If so, why for example would HTLV-1 screening cause more anxiety than other antenatal screening tests – eg for Down’s syndrome or HIV?

Sincerely,

Graham P Taylor, MB, DSc

Academic Editor

Eugenia Corrales-Aguilar

Section Editor

Thank you for addressing point by point the reviewers' comments. This has been most helpful. Further clarification of how onward sexual transmission was incorporated into the model is required as well as how the contribution of sexually acquired transmission to the total number of cases of HTLV-1 in Japan has been factored into the incidence data. See the points below and the further comments from Reviewer 1.

Line 131 – are 4000 infections in adults and adolescents diagnosed each year or is this the estimated number of new infections per year?

Line 217 and Table 1– the horizontal infection rate is 0.000046 on the 2016 paper of HTLV-1 incidence in Japanese blood donors. This is the infection rate but not the transmission rate. A carrier infected in early life will have potentially more opportunity to transmit than a carrier who does not become infected until they become sexually active. Please also confirm that the transmission rate when incorporated into the model describes the life-time risk of transmission following early life infection and not the risk per 100,000 person years. This is not clear at the moment. I note too that the work of Satake et al has recently been updated in J Clin Virol 2022.

(The incidence density was significantly higher in women (6·88 per 100 000 person-years; 95% CI 6·17-7·66) than in men (2·29 per 100 000 person-years; 95% CI 1·99-2·62; p<0·0001). 

The number of seroconversions per 100,000 person-years was 1.54 for men and 4.21 for women – satake et al J Clin Virol . 2022 Dec;157:105324. doi: 0.1016/j.jcv.2022.105324.

Although the seroconversions per 100,000 person years is lower than in the 2016 report the sero-conversion rate amongst adolescents and Young adults had increased.

Line 307 – “The sexual transmission rate didn’t affect the cost-effectiveness of screening”. Do you mean that the number of transmissions attributed to early life infection was so low that if did not increase the cost-effectiveness of screening. How many sexual transmissions would have been prevented by the avoidance of mother-to-child transmission. This should be added to the section on cumulative lifetime health effects.

In this section the estimate on ATL-associated deaths is 64% of the total of ATL cases whereas the data input was that the 4 year survival of 16%. Why is the mortality rate in the model so low. On the other hand 71 HAM associated deaths were prevented out of 177 cases – a 40% mortality. The suggestion in the text is that these numbers may be wrong. What is their contribution to the cost-effectiveness?

In the response to Reviewer one you state Yes, the association of ATL risk with acquisition of HTLV-1 in early life was considered in the analysis. In the model, HTLV-1 is transmitted mainly through mother-to-child transmission (MTCT), even if horizontal transmission through sexual intercourse is being considered. However, the data used for ATL risk was the published incidence rate – which would include all persons infected with HTLV-1 regardless of the age of infection. Whilst ATL has been reported following infection in adult life a strong association with infection in early life remains. The relative contribution of infection in early v adult life is therefore likely to impact significantly in the model. What proportion of all HTLV-1 infection in adults in Japan did you consider to be related to mother-to-child transmission? The increasing rates of infection with increasing age, especially among females, indicate a considerable infection after early life. 

In your response to point 16 Reviewer 1 you have stated that anxiety related to screening would be a particular feature of HTLV-1 screening. It is not clear that this was your intention. If not, please amend. If so, why for example would HTLV-1 screening cause more anxiety than other antenatal screening tests – eg for Down’s syndrome or HIV?

Reviewer's Responses to Questions

**Key Review Criteria Required for Acceptance?**

**Methods**

-Are the objectives of the study clearly articulated with a clear testable hypothesis stated?

-Is the study design appropriate to address the stated objectives?

-Is the population clearly described and appropriate for the hypothesis being tested?

-Is the sample size sufficient to ensure adequate power to address the hypothesis being tested?

-Were correct statistical analysis used to support conclusions?

-Are there concerns about ethical or regulatory requirements being met?

Reviewer #1: Methods are adequate.

Authors need to clarify if they have included secondary vertical transmissions, or horizontal transmission only (Page 13, line 217)

**Results**

-Does the analysis presented match the analysis plan?

-Are the results clearly and completely presented?

-Are the figures (Tables, Images) of sufficient quality for clarity?

Reviewer #1: Figure 1 A needs editing as it is not clear. Consider changing scale.

**Conclusions**

-Are the conclusions supported by the data presented?

-Are the limitations of analysis clearly described?

-Do the authors discuss how these data can be helpful to advance our understanding of the topic under study?

-Is public health relevance addressed?

Reviewer #1: Authors should keep both thresholds (50,000 and 100,000) in the discussion.

**Editorial and Data Presentation Modifications?**

Reviewer #1: The study is well designed, clear and important. I would strongly advise to keep both thresholds in the discussion 50,000 and 100,000. This information is important as it shows that HTLV antenatal screening is not only cot-effective, but highly cost-effective, as authors have used a conservative threshold.

**Summary and General Comments**

Reviewer #1: The paper is well written, timely and important. Only minor comments as above. The inclusion of short discussion adding both threshold is important in a broader context of cost-effectiveness of such intervention in other settings.

PLOS authors have the option to publish the peer review history of their article (what does this mean?). If published, this will include your full peer review and any attached files.

Reviewer #1: No

Figure Files:

Data Requirements:

Reproducibility:

References

---

## [Decision Letter · Decision Letter 2]

9 Jan 2023

Dear Dr. Kowada,

Thank you very much for submitting your manuscript "Cost-effectiveness and health impact of HTLV-1 antenatal screening for prevention of mother-to-child transmission" for consideration at PLOS Neglected Tropical Diseases. As with all papers reviewed by the journal, your manuscript was reviewed by members of the editorial board and by several independent reviewers. The reviewers appreciated the attention to an important topic. Based on the reviews, we are likely to accept this manuscript for publication, providing that you further modify the manuscript according to the review recommendations. 

The focus of concern is mainly (but not only) on the impact of horizontal transmission of HTLV-1 in the two scenarios and how the reduced transmission from the screened pregnant lady after diagnosis of HTLV-1 infection has been calculated based on the expected behaviour change as well as the horizontal transmission that does not occur during their lifetime from the infants in whom HTLV-1 was prevented. If either or both of these have not been included in the analysis, and there is insufficient data to include in the revision, then this should be made clear as a limitation and an underestimate of the potential benefit of the antenatal screening.

Please also address the other, minor points.

Sincerely,

Graham P Taylor, MB, DSc

Academic Editor

Eugenia Corrales-Aguilar

Section Editor

Reviewer's Responses to Questions

**Key Review Criteria Required for Acceptance?**

**Methods**

-Are the objectives of the study clearly articulated with a clear testable hypothesis stated?

-Is the study design appropriate to address the stated objectives?

-Is the population clearly described and appropriate for the hypothesis being tested?

-Is the sample size sufficient to ensure adequate power to address the hypothesis being tested?

-Were correct statistical analysis used to support conclusions?

-Are there concerns about ethical or regulatory requirements being met?

Reviewer #1: Line 166-167 author say that a hypothetical cohort of 30-year individuals was targeted. Is the author referring to pregnant women? The model should focus on pregnant women, since they are the target population of the intervention. This needs to be clear

Line 191 “Patients with favorable and smoldering types of ATL continue to be followed up without treatment”. How long are they followed up for? What rate of transformation to the more aggressive forms of ATL were imputed in the model.

ine 194-195 Author states "The probability of HTLV-1 transmission through sexual intercourse is considered", does it refer to the sexual transmission in the 30-year-old cohort (from sexual partners to pregnant women or from pregnant women to their sexual partners?) or the sexual transmission that may occur during the lifetime of the babies born from those mothers? If it refers to the 30-year-old cohort, it is not expected that sexual transmission would affect or be affected by antenatal screening program and should not be considered. However, prevention of sexual transmission to a partner can also be prevented by safer sex (condom use), once the women is aware of their serostatus due to antenatal screening. 

If it refers to secondary sexual transmission throughout the babies` life author should not use the incidence rate in blood donors. The horizontal transmission rate used is that of the incidence found among persons of reproductive age however all models of infection have shown the prevalence of infection to increase with age and that the rate of increase is higher in older age. Therefore the model should not use the same horizontal transmission rate throughout the person’s life as this under-estimates the number of infections prevented.

The transmission rate of HTLV through sex is expected to be much higher than incidence in blood donors at reproductive age, but would vary according to different variables, such as gender of the infected individual, number of partners, frequency of unprotected sex, co-infection with other STI, HTLV proviral load, etc. The rate of sexual transmission was estimated as 60% from male to female and 0.4% from female to male in a 10y period, but this is an estimation only and has a lot of uncertainties. In the Miyazaki cohort of sero-discordant couples the transmission rate over 5 years was 7% with a male-to-female rate 4-fold higher than females-to-males. Some references regarding sexual transmission rate

https://pubmed.ncbi.nlm.nih.gov/15809908/

https://pubmed.ncbi.nlm.nih.gov/8418183/

https://pubmed.ncbi.nlm.nih.gov/2877031/

In terms of those who are infected in early life – there is clearly a longer period of risk of horizontal transmission that those who are infected in later life, but we do not know whether this translates into a greater overall risk of horizontal transmission as behavioural factors may differ between the two risk groups.

If horizontal transmision is being considered the model should include the transmissions relating to the unscreened pregnant lady and transmissions relating to the infants who become infected as a result of the non-screening strategy. As there is only one row for horizontal transmission it seems that only the mother is considered but this is not completely clear.

**Results**

-Does the analysis presented match the analysis plan?

-Are the results clearly and completely presented?

-Are the figures (Tables, Images) of sufficient quality for clarity?

Reviewer #1: Line 304-305 "Cost effectiveness is not sensitive to the horizontal sexual transmission rate". See comment above.

Table 3. the cumulative lifetime HTLV-1 carriers with horizontal transmission is higher if antenatal screening is implemented, with a gain of 91 cases. This does not seem to be correct. Higher number of infected people is expected without antenatal screening, and therefore, more people would be able to transmit HTLV through sex. There is no reason that could explain higher number of sexual transmission if antenatal screening is implemented.

 Table 3. ATL cases are primarily associated with vertical transmission. Therefore, the reduction of number of ATL cases should be proportional to the reduction of number of infections that would be prevented. In table 3 number of carriers with vertical transmission reduces from 158,244 to 32,822, while number of ATL reduces from 14,804 to 10,392. Even considering horizontal transmission, the number of ATL seem to be too high. Please explain the model for generating the number of cases of ATL. It would appear from the numbers that the model considered most cases of ATL to be consequence of horizontal trasmission.

Table 3 – the LY gain is 456 despite the 4412 fewer cases of ATL – this amounts about 1 month of life expectancy gain for each case of ATL prevented – again this seems difficult to explain.

**Conclusions**

-Are the conclusions supported by the data presented?

-Are the limitations of analysis clearly described?

-Do the authors discuss how these data can be helpful to advance our understanding of the topic under study?

-Is public health relevance addressed?

Reviewer #1: (No Response)

**Editorial and Data Presentation Modifications?**

Reviewer #1: (No Response)

**Summary and General Comments**

Reviewer #1: The study addresses an important gap in the literature. It is not clear, however, how horizontal transmission was considered in the model. This is a major limitation and should be addressed.

PLOS authors have the option to publish the peer review history of their article (what does this mean?). If published, this will include your full peer review and any attached files.

Reviewer #1: No

Figure Files:

Data Requirements:

Reproducibility:

References

---

## [Editor Report · Decision Letter 3]

31 Jan 2023

Dear Dr. Kowada,

We are pleased to inform you that your manuscript 'Cost-effectiveness of HTLV-1 antenatal screening for prevention of mother-to-child transmission' has been provisionally accepted for publication in PLOS Neglected Tropical Diseases.

Best regards,

Graham P Taylor, MB, DSc

Academic Editor

Eugenia Corrales-Aguilar

Section Editor

---

## [Editor Report · Acceptance letter]

13 Feb 2023

Dear Dr. Kowada,

We are delighted to inform you that your manuscript, "Cost-effectiveness of human T-cell leukemia virus type 1 (HTLV-1) antenatal screening for prevention of mother-to-child transmission," has been formally accepted for publication in PLOS Neglected Tropical Diseases.

Best regards,

Shaden Kamhawi

co-Editor-in-Chief

Paul Brindley

co-Editor-in-Chief
